# Therapeutic Role of Polyphenol Extract from *Prunus cerasifera* Ehrhart on Non-Alcoholic Fatty Liver

**DOI:** 10.3390/plants13020288

**Published:** 2024-01-18

**Authors:** Jiabao Ren, Xing Zhang, SU Heiyan-Perhat, Po Yang, Helong Han, Yao Li, Jie Gao, Enpeng He, Yanhong Li

**Affiliations:** 1Key Laboratory of Special Environment Biodiversity Application and Regulation in Xinjiang, College of Life Sciences, Xinjiang Normal University, Urumqi 830054, China; rjb331154180@163.com (J.R.); zxyybh@163.com (X.Z.); 13579269571@163.com (S.H.-P.); hhl13779071291@163.com (H.H.); l2031021@163.com (Y.L.); jiegao72@gmail.com (J.G.); 2Key Laboratory of Sports Human Sciences, Institute of Physical Education, Xinjiang Normal University, Urumqi 830054, China; 15635295235@163.com; 3College of Arts and Sports, Hebei Institution of Communication College, Shijiazhuang 051430, China

**Keywords:** *Prunus cerasifera* Ehrhart, drought environments, polyphenol extracts, non-alcoholic fatty liver disease, bile acid metabolism

## Abstract

*Prunus cerasifera* Ehrhart (*P. cerasifera*) flourishes uniquely in the arid landscapes of Xinjiang, China. Preliminary studies have revealed the therapeutic potential of its polyphenol extract (PPE) in mitigating liver lipid accumulation in mice fed a high-fat diet. We established a mouse model that was subjected to a continuous high-fat diet for 24 weeks and administered PPE to investigate the effects of PPE on cholesterol and BA metabolism in NAFLD mice. The results showed that PPE administration (200 and 400 mg/kg/day, BW) led to a reduction in liver TC, an increase in liver T-BAs, and normalization of the disrupted fecal BA profile. Concurrently, it decreased levels of lipotoxic BAs and inhibited hepatic cholesterol synthesis (evidenced by reduced HMGCR activity) and intestinal cholesterol absorption (indicated by lower ACAT2 levels) while enhancing intestinal cholesterol efflux (via LXRα, ABCA1, ABCG5, and ABCG8) and stimulating hepatic BA synthesis (CYP7A1, CYP27A1) and secretion (BSEP). PPE thus led to a significant reduction in lipotoxic BAs metabolized by gut microbiota and a downregulation of the BA secretion pathway under its influence. Our findings reveal the therapeutic effect of PPE on NAFLD mice via regulating cholesterol and BA metabolism, providing a theoretical basis for exploring the potential functions of *P. cerasifera*.

## 1. Introduction

Plants from extreme regions, particularly those from arid areas, often possess significant biomedicinal properties. Non-alcoholic fatty liver disease (NAFLD), a prevalent global metabolic disorder, primarily arises from obesity. This condition is typified by an excessive accumulation of fat in the liver, a consequence of an imbalance where hepatic lipid synthesis surpasses lipid degradation [1]. Epidemiological studies indicate that NAFLD often represents the initial phase in the progression of more severe liver pathologies, including non-alcoholic steatohepatitis (NASH), liver fibrosis, and hepatocellular carcinoma (HCC). Despite its widespread nature and potential to advance to more serious diseases, effective clinical treatments specifically targeting NAFLD remain elusive, with no currently approved pharmaceutical interventions available [2,3]. This gap in treatment underscores the critical need for ongoing research and development in this area.

The understanding of NAFLD pathogenesis has evolved significantly, transitioning from the double-hit to the multiple-hit theory. This shift underscores the complexity of NAFLD, linking its onset to a network of molecular pathways within the body [4]. Central to these is the imbalance in cholesterol homeostasis, identified as a primary risk factor in NAFLD development [5]. Cholesterol homeostasis is a tightly regulated process, encompassing endogenous synthesis, exogenous uptake, efflux, and biotransformation. Endogenous cholesterol, primarily synthesized by hepatic HMG-CoA reductase (HMGCR), together with exogenous cholesterol, chiefly absorbed through the intestinal actions of NPC1-like intracellular cholesterol transporter 1 (NPC1L1) and acetyl-CoA acetyltransferase 2 (ACAT2), constitute the dominant sources of cholesterol intake [6]. The regulation of cholesterol efflux is mediated by the liver X receptor (LXRα), a pivotal member of the nuclear hormone receptor superfamily of ligand-dependent transcription factors. LXRα orchestrates this process by controlling the activity of ABC family transporters, such as ATP binding cassette subfamily A member 1 (ABCA1), ATP binding cassette subfamily G member 5 (ABCG5), and ATP binding cassette subfamily G member 8 (ABCG8), thereby maintaining cholesterol equilibrium [6,7]. Furthermore, a significant portion of cholesterol, approximately 40%, undergoes metabolic conversion into bile acids (BAs) as part of its degradation pathway [8]. This cholesterol-to-BA conversion is not just a degradation route but also a crucial regulatory mechanism for maintaining cholesterol homeostasis, underscoring the intricate balance and interplay of various biochemical pathways in the pathogenesis of NAFLD. Understanding these processes is essential for developing targeted therapeutic strategies for this increasingly prevalent disease.

Recent studies have illuminated a strong connection between bile acid (BA) metabolic disorders [9,10] and gut microbiota imbalance [11,12] in the context of non-alcoholic fatty liver disease (NAFLD). These findings underscore the complexity of NAFLD’s pathophysiology. Notably, the farnesoid X receptor (FXR), a crucial modulator of BAs homeostasis, plays a dual role in NAFLD’s progression [13]. In the liver, cholesterol undergoes transformation into primary BAs through two distinct pathways: the classical pathway involving cholesterol 7a-hydroxylase (CYP7A1) and the alternative pathway via sterol-27-hydroxylase (CYP27A1). These primary BAs then combine with taurine or glycine, forming conjugated BAs. These conjugated BAs are secreted into the intestine under the influence of dietary stimuli via the bile salt export pump (BSEP), playing a pivotal role in lipid digestion and absorption in the body. However, a disruption in this process, either by inhibited BAs synthesis or reduced BSEP activity (stemming from suppressed hepatic FXR-small heterodimer partner (SHP) axis signaling), exacerbates NAFLD [14,15,16]. Concurrently, NAFLD is often accompanied by a marked imbalance in gut microbiota. In the NAFLD context, there is a significant enrichment of microbiota with bile salt hydrolase (BSH) activity and dehydroxylation capabilities. This enrichment leads to an increased conversion of conjugated primary BAs into lipotoxic deconjugated secondary BAs. This conversion activates the signaling of the intestinal FXR-fibroblast growth factor 15 (FGF15) axis, further aggravating the progression of NAFLD [17,18,19]. Thus, the interplay between gut microbiota, BAs metabolism, and liver function emerges as a critical area in understanding and potentially managing NAFLD.

Wild *Prunus cerasifera* Ehrhart (*P. cerasifera*) is a valuable wild fruit resource distributed in the mountains of Central Asia. In China, it is only distributed in the narrow valleys of Daxigou and Xiaoxigou in the arid area of northwest China. The survey found that local residents who eat *P. cerasifera* fruits for a long time have a much lower rate of cardiovascular disease than other people. At the same time, *P. cerasifera* is a unique medicinal germplasm resource in Xinjiang, and its fruits are rich in polyphenols. These polyphenols, classified as plant secondary metabolites, have garnered significant interest for their efficacy in combating metabolic diseases. For instance, various polyphenolic compounds, such as those found in apples, alongside berberine, hyperoside, and elexosaponin A1, have demonstrated considerable potential in mitigating the progression of non-alcoholic fatty liver disease (NAFLD) in animal models. This therapeutic effect is attributed to their ability to modulate cholesterol metabolism, bile acids (BAs) metabolism, and gut microbiota [13,20,21,22]. These findings not only highlight the therapeutic promise of polyphenols in metabolic disorders but also underscore the intricate biological mechanisms through which these compounds exert their beneficial effects.

In our previous research, we found that the polyphenol extract from *P. cerasifera* (PPE) consists of 18 polyphenol monomers, including 3-O-caffeoylquinic acid, 5-O-caffeoylquinic acid, 4-O-caffeoylquinic acid, and Apigenin-O-glucoside (refer to Appendix A). Furthermore, this extract has been shown to have a notable therapeutic effect on obesity [23,24]. Despite these promising findings, it remains uncertain if PPE can effectively mitigate the development of non-alcoholic fatty liver disease (NAFLD) through the modulation of cholesterol and bile acids (BAs) metabolism. This area of uncertainty presents a crucial avenue for further exploration to understand the full potential of PPE in treating metabolic disorders. Addressing the aforementioned uncertainties, we have formulated two key research questions to guide our investigation: (1) Does PPE exert a significant impact on cholesterol and bile acids (BAs) metabolism in NAFLD mouse models? (2) What are the specific mechanistic pathways through which PPE modulates cholesterol and BAs metabolism? These queries are fundamental in advancing our understanding of PPE’s potential role in the management of NAFLD.

## 2. Results

### 2.1. Effects of PPE on TC, T-BAs, and Liver Damage in the Liver

To explore whether PPE is effective in NAFLD, different doses of PPE were administered to mice with NAFLD induced by long-term high-fat feeding. The concentrations of total cholesterol (TC) and total bile acids (T-BAs) in the liver are depicted in Figure 1A,B. The TC and T-BAs in the HFD group were higher than those in the CK group, which were 12.06 ± 2.35 mmol/g and 64.15 ± 10.63 μmol/L, respectively. Remarkably, upon PPE intervention, the hepatic TC levels were substantially reduced, while the T-BAs levels were significantly elevated in comparison to those observed in the HFD group. This indicates a notable effect of PPE on modulating these key metabolic parameters.

To ascertain the protective impact of PPE on liver and adipose tissues, sections of both liver and fat were prepared (Figure 1C). Notably, the NAFLD activity score in the liver, encompassing parameters like inflammation, ballooning, and steatosis, along with the size of adipocytes in NAFLD mice, exhibited a significant reduction following PPE intervention. This is evidenced in the detailed analyses presented in Figure 1D,E, highlighting the effectiveness of PPE in mitigating the key pathological features associated with NAFLD.

### 2.2. Effects of PPE on Cholesterol Metabolism

HMG-CoA reductase (*HMGCR*) mRNA levels were significantly higher in the high-fat diet (HFD) group compared to the control (CK), but they notably decreased in the high-dose HFD (HFD+H) group. Liver X receptor alpha (*LXRα*) mRNA showed a similar trend, with a lower expression in the HFD group and a significant increase in the HFD+H group. In the ileum, expressions of ATP binding cassette transporters A1 (*ABCA1*), G5 (*ABCG5*), G8 (*ABCG8*), and Niemann–Pick C1-Like 1 (*NPC1L1*) were significantly reduced in the HFD group but increased in the HFD+H group. Conversely, acetyl-CoA acetyltransferase 2 (*ACAT2*) mRNA was higher in the HFD group but decreased in the HFD+H group. These differential gene expressions in response to diet and treatment are detailed in Figure 2A.

The expression of HMGCR and LXRα proteins in the HFD group was notably higher than in the CK group. However, in the HFD+H group, HMGCR protein expression significantly decreased, while LXRα expression increased in comparison to the HFD group, as detailed in Figure 2B,C.

### 2.3. Effects of PPE on Fecal BAs Profile in Mice

In the high-fat diet (HFD) group, fecal total bile acids (T-BAs) and other BA proportions were significantly elevated compared to the control (CK) group, with decreased unconjugated and glycine-conjugated BAs. PPE intervention notably reversed these alterations (Figure 3A,B). Additionally, the HFD group showed a lower secondary/primary BA ratio than the CK group. Concentrations of primary BAs (cholic acid (CA), chenodeoxycholic acid (CDCA), β-cholic acid (βCA), β-muricholic acid (βMCA), ω-muricholic acid (ωMCA), cholic acid-3-sulfate (CA-3S)) and secondary BAs (deoxycholic acid (DCA), 23-nordeoxycholic acid (23norDCA), lithocholic acid (LCA)) were higher in the HFD group. PPE intervention significantly adjusted the primary and secondary BAs concentrations and their ratio (Figure 3C,D).

### 2.4. Effects of PPE on BAs Metabolism

Gene Expression in BAs Metabolism (Figure 4A): In the HFD group, hepatic *CYP27A1* and *BSEP* showed no significant changes compared to CK, but the expressions of *CYP7A1*, *FXR*, *SHP*, and Na+/taurocholate cotransporter (*NTCP*) were markedly reduced. Compared with the HFD group, the HFD+H group exhibited increased levels of liver *CYP7A1*, *CYP27A1*, *FXR*, *SHP*, *BSEP,* and *NTCP*. Ileal Gene Expression: Compared to CK, the HFD group had higher ileal *FXR* and lower Apical sodium-dependent bile acid transporter (*ASBT*) expressions, with no effect on *FGF15*. PPE notably downregulated *FXR* and *FGF15* and upregulated *ASBT* in the ileum relative to HFD (Figure 4A). Protein Expression in BAs Metabolism (Figure 4B,C): In the liver, the HFD+H group significantly increased FXR, SHP, and CYP7A1 expressions compared to HFD. Post-PPE, the ileal ASBT expression decreased notably compared to the HFD group.

### 2.5. Correlation Analysis between Gut Microbiota and BAs Metabolism

The above studies have confirmed that PPE alleviates NAFLD by regulating cholesterol metabolism and BAs metabolism, but it remains unclear whether BAs metabolism is related to PPE-driven structural changes in gut microbiota. Therefore, a correlation analysis was conducted on the structural changes in gut microbiota and changes in BAs metabolism in NAFLD mice after PPE intervention. At the genus level, Spearman correlation analysis was performed on the top 30 differential microorganisms and 15 differential BAs between the HFD and HFD+H groups, and data without significant correlation were filtered out. There were negative correlations between taurohyodeoxycholic acid (THDCA) and isohyodeoxycholic acid (isoHDCA) with Mucispirillum, Erysipelatoclostridium, and Bifidobacterium and positive correlations with Lactobacillus, Akkermansia, Parabacteroides, Neglecta, Turicibacter, Lachnospiraceae_NK4A136_group, Monoglobus, Candidatus_Saccharimonas, and Clostridia_UCG−014_unclassified (Figure 5A). Gut microbiota with BSH activity (Turicibacter, Streptococcus, Lactobacillus, Alloprevotella and Bacteroides) were enriched in the HFD+H group compared to the HFD group (Figure 5B,C), but the ratio of DCA/(DCA+CA) was significantly reduced (Figure 5D).

### 2.6. Effects of PPE on Serum Metabolic Profile in Mice

The OPLS-DA model indicated significant metabolic differences between the HFD and HFD+H groups, with a total of 179 differential metabolites identified in HFD vs. HFD+H (Figure 6A and Appendix A). Among these differential metabolites, there were 10 significantly upregulated, 10 significantly downregulated, and 159 insignificantly differential metabolites (Figure 6B). Notably, cholic acid was among the 10 significantly downregulated differential metabolites in the HFD+H group. Pathway enrichment analysis demonstrated a significant enrichment of the bile secretion pathway, with a proportion of 7.27% of detected metabolites participating in this pathway (Figure 6C–E).

## 3. Discussion

NAFLD is considered an epidemic affecting approximately 1/4 of the world’s population, and its pathogenesis is complex. Currently, research on this disease is mainly conducted using mouse models. Hepatic lipid accumulation plays a pivotal role in the pathogenesis of NAFLD, contributing significantly to the development and progression of the disease [5]. Our experimental findings provide compelling evidence that PPE treatment effectively mitigated the excessive hepatic accumulation of TC, a hallmark of NAFLD, in the murine model. Importantly, these observations were further substantiated by histopathological analyses, which unequivocally demonstrated the hepatoprotective properties of PPE in NAFLD-afflicted mice. The histopathological assessments revealed a notable reduction in hepatic steatosis, inflammation, ballooning, and overall NAFLD activity score in the PPE-treated group when compared to untreated NAFLD mice. These histological improvements underscore the potential therapeutic efficacy of PPE in ameliorating the liver damage associated with NAFLD.

Numerous studies have provided compelling evidence that the metabolism of bile acids (BAs) not only contributes to the development of NAFLD [9,10,13,25] but also represents a promising target for NAFLD treatment [26]. In the context of high-fat-induced NAFLD in mice, there is a noticeable tendency for hepatic total BAs (T-BAs) to increase, accompanied by a significant rise in fecal T-BAs. This suggests that the rate of lipid degradation is constrained in NAFLD [1]. Intervention with PPE leads to a decrease in fecal T-BAs and an increase in liver T-BAs, indicating that the intervention facilitates the conversion of cholesterol into BAs and their excretion into the intestine. To substantiate this hypothesis, we further investigated the activity of CYP7A1 and CYP27A1, the rate-limiting enzymes responsible for the synthesis of BAs from cholesterol, particularly cholic acid (CA) and chenodeoxycholic acid (CDCA), in the liver through the classical and alternative pathways [10]. Our findings demonstrate that NAFLD mice do convert a portion of cholesterol into BAs via CYP7A1 and CYP27A1, but the rate of lipid degradation is considerably lower than that observed with a high-dose PPE intervention. Of particular interest is the conversion of most CDCA into α-/β-murine cholic acid (α-/β-MCA) through 6α-/β-hydroxylation in rodents [27]. Among these, CA, CDCA, and α-/β-MCA are conjugated with taurine or glycine in the liver to form conjugated BAs, which then flow into the gallbladder. Subsequently, these conjugated BAs are discharged into the intestine via the bile salt export pump (BSEP) under the coordination of the FXR-SHP axis to participate in the digestion process [13,14]. At both the gene and protein levels, PPE intervention led to a significant upregulation in the expression of liver FXR, SHP, and BSEP. However, low-dose PPE had no discernible effect on FXR. These data collectively suggest that PPE enhances BAs synthesis in the liver and promotes BAs efflux by activating the FXR-SHP axis signaling pathway. Nevertheless, it is noteworthy that serum metabolic profiles predicted a downregulation of the bile secretion pathway, while liver BSEP was upregulated. This discrepancy might be attributed to inconsistencies in gene and protein expression patterns and warrants further investigation.

In contrast to the activation of hepatic FXR, the activation of intestinal FXR in NAFLD mice has been associated with the promotion of NAFLD through the FXR-FGF15 signaling pathway [17,18,19]. However, emerging evidence suggests that polyphenols exert a downregulatory effect on intestinal FXR-FGF15 signaling by modulating the composition of gut microbiota, particularly those involved in bile acid metabolism, in the context of metabolic diseases [28,29,30]. Our study yielded intriguing findings, revealing that PPE intervention effectively attenuated ileal FXR-FGF15 signaling in NAFLD-afflicted mice. Notably, the inhibitory effect was more pronounced in the high-dose PPE intervention group. This observation sheds light on a potentially novel mechanism by which PPE may exert its therapeutic effects in NAFLD. It appears that PPE’s influence on NAFLD extends beyond hepatic factors as it impacts intestinal FXR-FGF15 signaling, offering a multifaceted approach to mitigate the development and progression of this complex metabolic disorder. Further investigations into the precise mechanisms underlying this effect are warranted and could hold significant implications for NAFLD treatment strategies.

Bile acid biosynthesis serves as a pivotal pathway for cholesterol efflux, complementing the other pathway primarily mediated by the ileal ABC family under the regulatory influence of hepatic LXRα. Following PPE intervention, there was a substantial increase in the expression of hepatic LXRα and ileal ABCA1, ABCG5, and ABCG8 at both gene and protein levels. These findings provide compelling evidence that PPE effectively facilitates cholesterol efflux from the liver. Excessive cholesterol levels have been shown to inhibit the functionality of ileal NPC1L1, thereby promoting cholesterol uptake by ileal ACAT2 [6,31]. Remarkably, PPE intervention resulted in the reversal of this cholesterol uptake mechanism. HMGCR, a key rate-limiting enzyme responsible for endogenous cholesterol synthesis in the liver, is known to be competitively inhibited by statins [32]. Our study unequivocally confirmed that PPE exerts an inhibitory effect on the expression of HMGCR. These insights into the regulatory effects of PPE on key players in cholesterol metabolism shed light on its potential therapeutic role in mitigating cholesterol-related disorders.

Alterations in the composition of gut microbiota can exert a profound impact on the fecal BAs profile and, conversely, changes in the fecal BAs profile can reciprocally influence gut microbiota. These intricate interactions between gut microbiota and the fecal BAs profile play a pivotal role in regulating host health through enterohepatic axis circulation [11,12,19]. In our study, we observed noteworthy changes in the fecal BAs profile of NAFLD mice following PPE intervention. Specifically, there was a significant increase in the low-level secondary/primary BA ratio and the proportion of unconjugated BAs, while the DCA/(DCA + CA) ratio exhibited a significant decrease. This phenomenon can be attributed to the remarkable enrichment of gut microbiota with bile salt hydrolase (BSH) activity, including Lactobacillus [33] and Turicibacter [34], in the intestine upon PPE intervention. These microbiota play a crucial role in converting conjugated primary BAs into deconjugated primary BAs, which are subsequently transformed into secondary BAs through 7-α-dehydroxylation. Notably, secondary BAs such as DCA and LCA have been linked to the development of various metabolic disorders [19]. In the context of NAFLD, DCA and LCA further exacerbate the progression of the disease by activating intestinal FXR-FGF15 axis signaling [17,18,35]. The reduction in elevated levels of DCA and LCA in the intestines of NAFLD mice following PPE intervention is indicative of a potential mechanism through which PPE inhibits the development of NAFLD. To validate the intricate interplay between fecal BAs and gut microbiota, Spearman correlation analysis was conducted. The results revealed significant negative correlations between DCA and LCA in feces and certain gut microbiota species, including Turicibacter [34] and Neglecta, while positive correlations were observed with others such as Mucispirillum [36,37] and Erysipelatoclostridium [38]. Additionally, secondary BAs THDCA and isoHDCA exhibited significant enrichment [39,40] following PPE intervention and displayed positive correlations with Turicibacter, Lactobacillus, Candidatus_Saccharimonas, Parabacteroides, Akkermansia, and Neglecta. These findings provide robust evidence that PPE alleviates NAFLD in mice by reshaping gut microbiota composition, reducing lipotoxic secondary BAs (DCA and LCA), and enhancing THDCA and isoHDCA levels to suppress intestinal FXR-FGF15 axis signaling.

## 4. Materials and Methods

### 4.1. Preparation of PPE

In the same batch, we harvested a sufficient amount of fresh wild *P. cerasifera* fruits in Yili, Xinjiang, China, on 20 August 2022 and immediately quick-froze them to −20 °C. We made slight modifications to the PPE extraction process based on previously reported methods [41]. The process involved the following steps: First, fresh *P. cerasifera* fruits were carefully deseeded and homogenized. Subsequently, the homogenized fruit material was subjected to extraction using a mixture of methanol, water, and formic acid (in a ratio of 90:9:1, *v*/*v*) at room temperature for a duration of 4 h. This extraction process was repeated twice to ensure thorough extraction. The resulting extract solution underwent a purification step using AB-8 macroporous adsorption resin. After complete adsorption onto the resin, the extraction solution was meticulously washed with a solution containing 1% formic acid. Subsequently, the target compounds were effectively eluted from the resin using 60% ethanol. The eluent obtained from this process was carefully collected and subjected to concentration and freeze-drying, ultimately yielding a fine powder. The total polyphenol content of *P. cerasifera* polyphenol extract was 439.17 mg/g, and the yield after purification was 68.54%.

### 4.2. Animals Experimental Design

A total of fifty-six 4-week-old ICR mice were procured from Xinjiang Medical University in Xinjiang, China. The mice underwent an initial period of standardized adaptive feeding, which involved maintaining them in controlled environmental conditions with a temperature of 25 ± 2 °C, relative humidity of 60 ± 5%, and a 12 h light and dark cycle. During this phase, the mice had unrestricted access to both food and water.

Following one week of adaptive feeding, the mice were randomly assigned to two main groups: a control group (referred to as the CK group, consisting of *n* = 13 mice) that was fed a standard chow diet and a high-fat group (referred to as the HFD group, consisting of *n* = 43 mice) that received a high-fat (60 FDC) purified rodent diet (HF60, Dyets Biotechnology, Wuxi, China). After a period of 11 weeks on their respective diets, three mice from each group were selected for assessment to confirm the successful establishment of the NAFLD model. This confirmation was achieved by evaluating serum lipid levels and assessing liver damage.

Subsequently, the NAFLD-afflicted mice were randomly divided into four distinct groups, as outlined in Figure 7, and subjected to an additional 24 weeks of intervention. The group divisions were as follows: (1) High-fat diet + normal saline (oral administration of equal volume of normal saline, HFD group, *n* = 10), (2) high-fat diet + high-dose PPE (oral administration, 400 mg/kg/day, BW, HFD+H group, *n* = 10), (3) high-fat diet + low-dose PPE (oral administration, 200 mg/kg/day, BW, HFD+L group, *n* = 10), and (4) high-fat diet + statin (oral administration, 10 mg/kg/day, BW, HFD+S group, *n* = 10).

Prior to the conclusion of the intervention period, fecal samples were collected for subsequent analysis of the targeted bile acids profile. At the conclusion of the experiment, all mice were euthanized via cervical dislocation. Various biological samples, including blood, adipose tissue, liver, and ileum, were harvested for further analysis.

The study was conducted in accordance with the Declaration of Helsinki and approved by the Animal Experimental Ethical Inspection of College of Life Sciences, Xinjiang Normal University (No. XJNU2022-08).

### 4.3. Liver Total Cholesterol (TC) and Total Bile Acids (T-BAs) Determination

The concentration of TC and T-BAs in the liver was measured following the recommended instructions provided by the kit manufacturer (Nanjing Jiancheng Bioengineering Institute, Nanjing, China). Liver tissue homogenates were subjected to centrifugation at 4 °C, 4000 rpm for 5 min, and the resulting supernatant was collected. Subsequently, all the collected samples were assayed in accordance with the specifications outlined by the kit.

### 4.4. Histopathological Observation of the Liver and Fat

Following the previously established procedure, liver and adipose tissues were procured and subsequently fixed using 4% paraformaldehyde. These tissues underwent dehydration using ethanol, were embedded in paraffin, sliced into sections, and ultimately stained with hematoxylin and eosin (H&E). The alterations in liver and adipose tissues were examined under a light microscope, enabling a visual assessment of the extent of liver lesions and the dimensions of adipocytes. Details of NAS semi-quantitative scoring are provided in Appendix A.

### 4.5. Determination of Targeted BAs Metabolism Profile in Feces and Non-Targeted Metabolic Profile in Serum

Mouse feces and serum samples, collected under aseptic conditions, were rapidly frozen using liquid nitrogen. Subsequently, these frozen samples were dispatched to Shanghai biotree Biomedical Technology Co., Ltd. (Shanghai, China) for comprehensive targeted and non-targeted metabolic profile analysis. The fecal and serum supernatants, prepared for analysis, were subjected to identification utilizing UHPLC-MS/MS. The raw data obtained were meticulously filtered and subsequently employed for the quantitative assessment of the fecal bile acids (BAs) profile and the serum metabolic profile. For further insights into the methodologies employed, please refer to the Appendix A section.

### 4.6. Real-Time PCR Analysis

Liver and ileum tissue samples were removed from a −80 °C refrigerator, and the total RNA was extracted using Trizol reagent; the RNA was reverse-transcribed to cDNA using a cDNA synthesis kit (Tiangen Biochemical Technology, Beijing, China). Then, the cDNA was used to amplify the target genes using SYBR Green PCR Master Mix on a StepOnePlus Real-time PCR detection system (4376600, ABI StepOnePlus, Carlsbad, CA, USA). The relative transcription of mRNA was calculated by the 2^−ΔΔCt^ method with β-actin for normalization, and the primer sequences are shown in Appendix A.

### 4.7. Western Blot Analysis

The liver and ileum tissue homogenates were treated with a lysis buffer containing protease inhibitors. Following homogenization, the resulting supernatant was separated by centrifugation, and the protein concentration was determined using the BCA protein assay kit from Nanjing Jiancheng Bio-Engineering Institute, Nanjing, China. Subsequently, the proteins were resolved through 12% SDS-PAGE, transferred onto PVDF membranes, and subjected to overnight incubation at 4 °C with primary antibodies targeting proteins such as CYP7A1, SHP, HMGCR, CYP27A1, LXR-α, FXR, ASBT, and GAPDH. The membranes were then washed thrice with TBST, followed by incubation with secondary antibodies for 2 h at room temperature, and then subsequent washing with TBST took place. Visualization of the blots was achieved using the Clarity Western enhanced chemiluminescence (ECL) substrate kit (P1050, Applygen Technologies, Beijing, China). Band density values were quantified and analyzed using Image J2x 2.1.5.0 software.

### 4.8. Statistical Analysis

Statistical analysis was conducted using SPSS 21.0 software, and the experimental data were expressed as mean ± SD, providing a comprehensive representation of the results. To evaluate differences between groups, an ANOVA Dunnett’s multiple-comparison test was employed, with statistical significance set at *P* < 0.05. The graphical illustrations were created using either GraphPad Prism 8.0 software or R Studio (R version 4.0), ensuring a visually clear and informative presentation of the findings.

## 5. Conclusions

This study has effectively demonstrated that PPE exerts a pronounced inhibitory effect on liver cholesterol biosynthesis and intestinal cholesterol absorption. This results in an increased conversion of liver cholesterol into bile acids (BAs) and enhanced intestinal cholesterol efflux. Furthermore, PPE attenuates the transformation of conjugated primary BAs into lipotoxic BAs, including LCA and DCA. These findings strongly suggest that PPE has the potential to alleviate NAFLD by modulating cholesterol and BAs metabolism in mice, as illustrated in Figure 8. This research lays the groundwork for considering *P. cerasifera* fruit as a functional food for NAFLD alleviation. However, the precise mechanisms underlying the alleviating effects of PPE intervention on NAFLD warrant further in-depth exploration.

## Figures and Tables

**Figure 1 plants-13-00288-f001:**
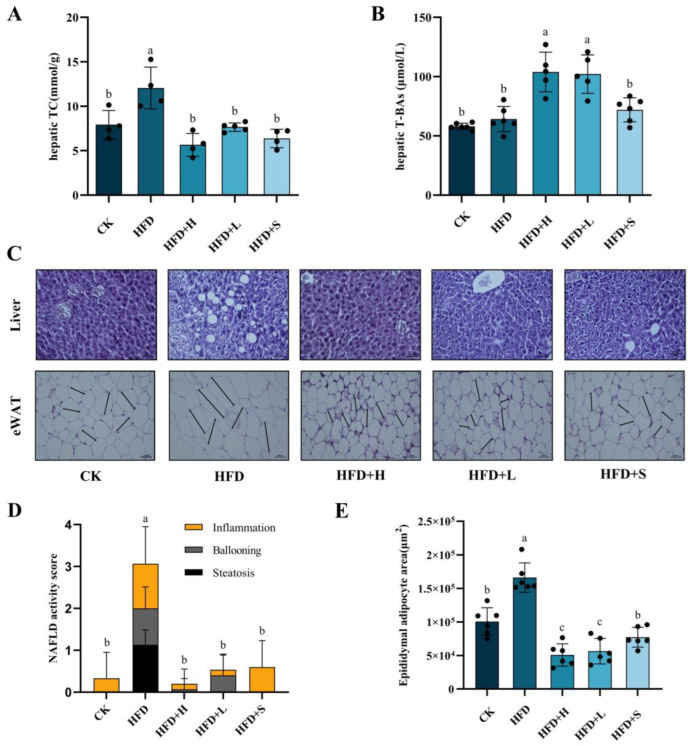
The levels of total cholesterol (TC), total bile acids (T-BAs), and indicators of liver damage were meticulously assessed. Panels (**A**,**B**) illustrate the concentrations of hepatic TC and T-BAs, respectively. Panel (**C**) displays representative histological images of liver and fat sections, stained with hematoxylin and eosin (H&E) at 200× magnification and the scale bar is 200 μm. The NAFLD Activity Score (NAS) of liver sections is depicted in panel (**D**), while panel (**E**) presents an analysis of adipocyte size within fat sections. The data are presented as mean ± SD. Statistically significant differences among the multiple groups (*P* < 0.05) are denoted by distinct letters, highlighting the differential impacts of these treatments on liver health and adiposity. CK: mice fed a standard chow diet with normal saline. HFD: mice fed a high-fat diet with normal saline. HFD+H: mice fed a high-fat diet with high-dose PPE (400 mg/kg/day). HFD+L: mice fed a high-fat diet with low-dose PPE (200 mg/kg/day). HFD+S: mice fed a high-fat diet with statin (10 mg/kg/day).

**Figure 2 plants-13-00288-f002:**
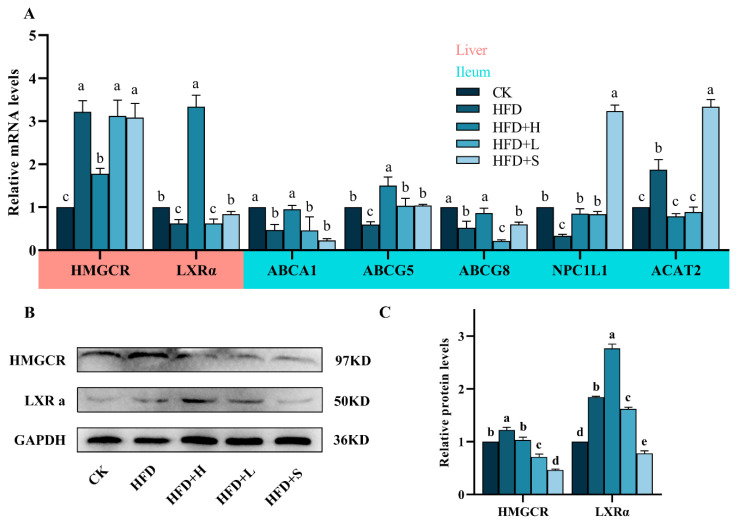
Impact of PPE on cholesterol metabolism in mice: (**A**) mRNA levels of key cholesterol metabolism genes (*HMGCR, LXR, ABCA1, ABCG5, ABCG8, NPC1L1, ACAT2*) in liver and ileum. (**B**) Western blot analysis of HMGCR and LXRα protein expression in liver. (**C**) Quantitative results of protein bands. Data are shown as mean ± SD (*n* = 3), with different letters indicating significant differences among groups (*P* < 0.05). CK: mice fed a standard chow diet with normal saline. HFD: mice fed a high-fat diet with normal saline. HFD+H: mice fed a high-fat diet with high-dose PPE (400 mg/kg/day). HFD+L: mice fed a high-fat diet with low-dose PPE (200 mg/kg/day). HFD+S: mice fed a high-fat diet with statin (10 mg/kg/day).

**Figure 3 plants-13-00288-f003:**
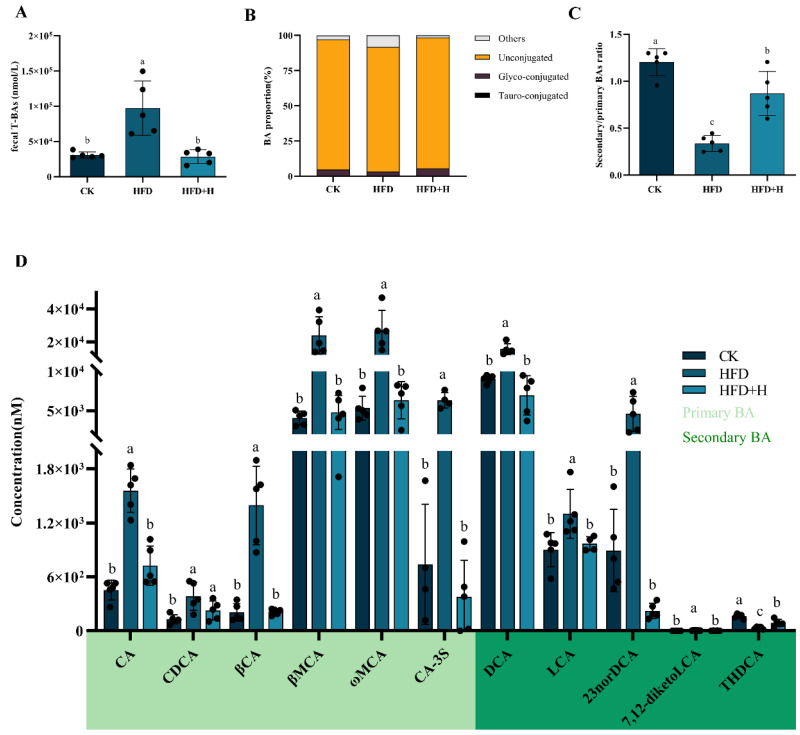
PPE’s impact on mice fecal BAs profile: utilizing UHPLC-MS/MS for targeted metabolomics, we analyzed mouse feces to assess (**A**) fecal total bile acid (T-BA) concentrations, (**B**) ratios of conjugated and unconjugated BAs, (**C**) secondary/primary BAs ratio, and (**D**) levels of various primary and secondary BAs. Data are presented as mean ± SD (*n* = 5), with different letters indicating significant differences across groups (*p* < 0.05). CK: mice fed a standard chow diet with normal saline. HFD: mice fed a high-fat diet with normal saline. HFD+H: mice fed a high-fat diet with high-dose PPE (400 mg/kg/day). HFD+L: mice fed a high-fat diet with low-dose PPE (200 mg/kg/day). HFD+S: mice fed a high-fat diet with statin (10 mg/kg/day). CA: cholic acid. CDCA: chenodeoxycholic acid. βCA: β-cholic acid. βMCA: β-muricholic acid. ωMCA: ω-muricholic acid. CA-3S: cholic acid-3-sulfate. DCA: deoxycholic acid. LCA: lithocholic acid. 23norDCA: 23-nordeoxycholic acid. 7,12-diketoLCA: 7,12-diketolithocholic acid. THDCA: taurohyodeoxycholic acid.

**Figure 4 plants-13-00288-f004:**
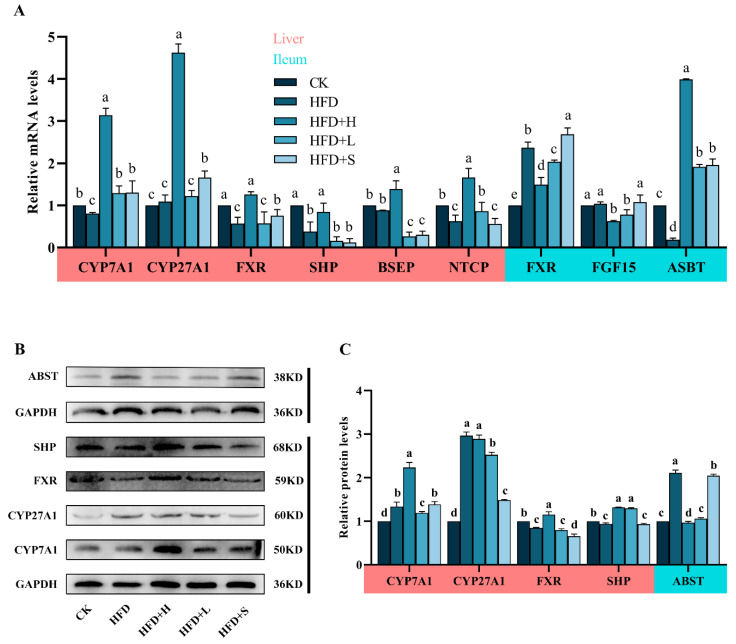
Impact of PPE on BAs metabolism in mice: (**A**) mRNA levels of crucial BAs metabolism genes (*CYP7A1, CYP27A1, FXR, SHP, BSEP, NTCP, FGF15, ASBT*) in liver and ileum. (**B**) Western blot analysis of ABST, SHP, FXR, CYP27A1, and CYP7A1 proteins in liver and ileum. (**C**) Quantitative results of protein bands. Data are shown as mean ± SD (*n* = 3), with different letters indicating significant differences across groups (*P* < 0.05). CK: mice fed a standard chow diet with normal saline. HFD: mice fed a high-fat diet with normal saline. HFD+H: mice fed a high-fat diet with high-dose PPE (400 mg/kg/day). HFD+L: mice fed a high-fat diet with low-dose PPE (200 mg/kg/day). HFD+S: mice fed a high-fat diet with statin (10 mg/kg/day).

**Figure 5 plants-13-00288-f005:**
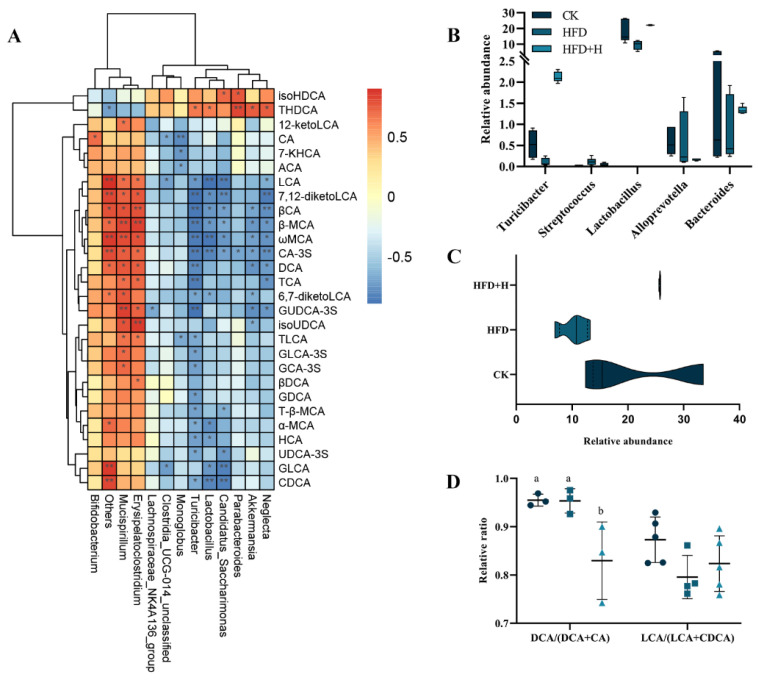
Correlation analysis between gut microbiota and BAs metabolism. (**A**) Spearman correlation analysis of differential gut microbiota and differential BAs at the genus level. (**B**) Differential BSH-producing gut microbiota at the genus level. (**C**) Sum of BSH-producing gut microbiota at the genus level. (**D**) DCA/(DCA + CA) and LCA/(LCA + CDCA) at the genus level. Data are presented as the mean ± SD (*n* = 5). Different letters represent a significant difference among multiple groups (*P* < 0.05) and * *P* ≤ 0.05, ** *P* ≤ 0.01. CK: mice fed a standard chow diet with normal saline. HFD: mice fed a high-fat diet with normal saline. HFD+H: mice fed a high-fat diet with high-dose PPE (400 mg/kg/day).

**Figure 6 plants-13-00288-f006:**
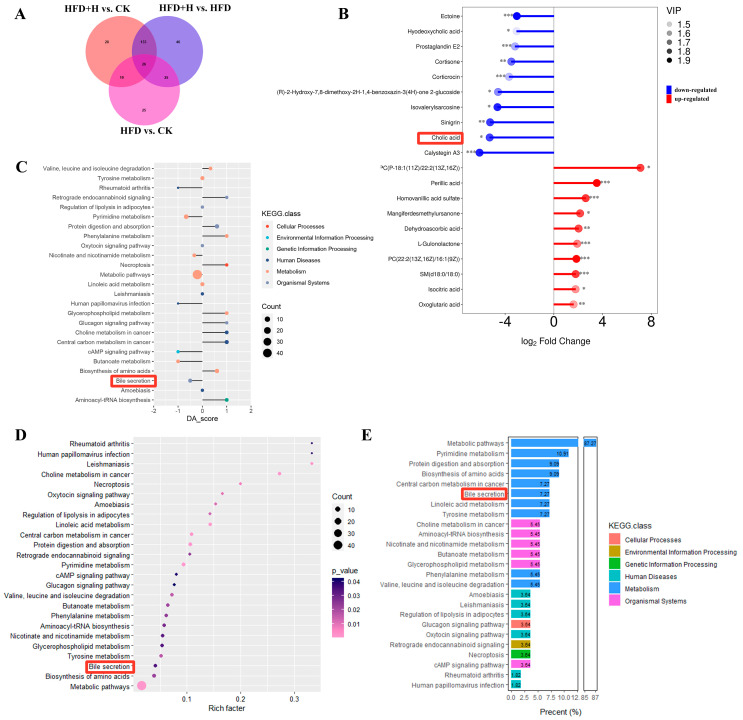
Effects of PPE on serum metabolic profile in mice. (**A**) Veen plot of differential metabolites. (**B**) Matchstick plot of differential metabolites. (**C**) KEGG pathway enrichment analysis matchstick plot. (**D**) KEGG pathway enrichment analysis bubble chart. (**E**) Detected metabolite proportion histogram. * *P* ≤ 0.05, ** *P* ≤ 0.01, *** *P* ≤ 0.001. (**B**–**E**) red frames: metabolites and pathways related to BA in serum metabolic profile. CK: mice fed a standard chow diet with normal saline. HFD: mice fed a high-fat diet with normal saline. HFD+H: mice fed a high-fat diet with high-dose PPE (400 mg/kg/day).

**Figure 7 plants-13-00288-f007:**
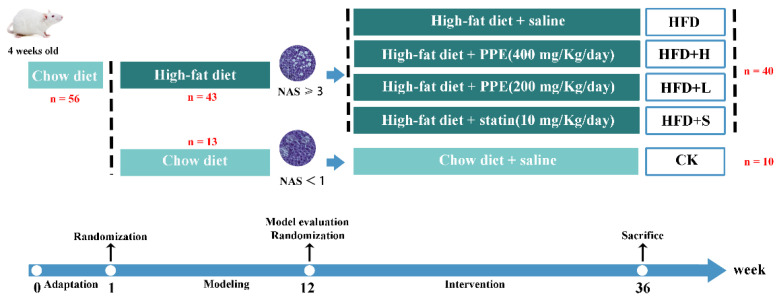
Animal experimental design. PPE: *P. cerasifera* polyphenol extracts. CK: mice fed a standard chow diet with normal saline. HFD: mice fed a high-fat diet with normal saline. HFD+H: mice fed a high-fat diet with high-dose PPE (400 mg/kg/day). HFD+L: mice fed a high-fat diet with low-dose PPE (200 mg/kg/day). HFD+S: mice fed a high-fat diet with statin (10 mg/kg/day). NAS: NAFLD activity score.

**Figure 8 plants-13-00288-f008:**
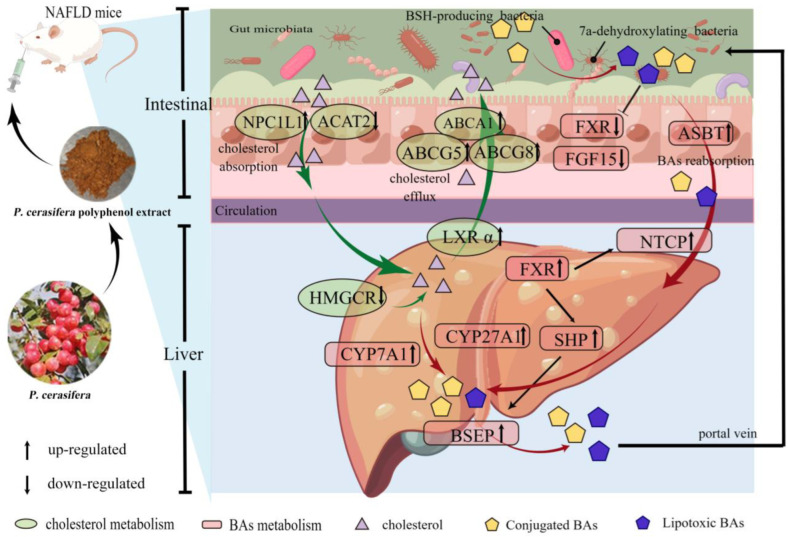
Potential molecular mechanisms of PPE affecting NAFLD. HMGCR: HMG-CoA reductase. NPC1L1: NPC1-like intracellular cholesterol transporter 1. ACAT2: acetyl-CoA acetyltransferase 2. LXRα: liver X receptor. ABCA1/5/8: ATP binding cassette subfamily A member 1/5/8. FXR: farnesoid X receptor. CYP7A1: cholesterol 7a-hydroxylase. CYP27A1: sterol-27-hydroxylase. BSEP: bile salt export pump. SHP: small heterodimer partner. FGF15: fibroblast growth factor 15. ASBT: apical sodium-dependent bile acid transporter. NTCP: Na+/taurocholate cotransporter. BAs: bile acids.

## Data Availability

Data are available in a publicly accessible repository.

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
