# Peer review of "Therapeutic Role of Polyphenol Extract from Prunus cerasifera Ehrhart on Non-Alcoholic Fatty Liver"

_plants, 2024, doi:10.3390/plants13020288_

Round 1
Reviewer 1 Report
Comments and Suggestions for Authors
The paper presents interesting study concerning activity of polyphenolic fraction derived from Prunus cerasifera in NAFLD. I have a few suggestions that might be helpful in improving the article.
1. The extracts used in the study are not characterized. We do not know anything on the ingredients. It is well known that extracts gained from one species can differ depending on many conditions - time of harvesting, drying, method of extraction.
2. Furthemore, it is not presented what extraction method was utilized.
3. In the introduction part a few words more could be said on the plant that was used in the study. It is not clear why it was chosen (except for a statement in the abstract that preliminary studies were promising).
4. Generally, the presentation of the results is clear but I would suggest to add in the description below the figures what HFD, HFD+H etc. mean. It is clear after reading the article but the figures should be readable independently from the text in the manuscript.
Good luck!
Reviewer 2 Report
Comments and Suggestions for Authors
Comments to the Author and Editor plants-2789055
The authors have explored the role of Prunus cerasifera Ehrh. polyphenol extracts in non-alcoholic fatty liver disease analysing the mechanism of action. During review, the following concerns arose:
1. In the Results Section 2.1. Effects of PPE on TC, T-BAs and liver damage in the live, the authors should give a short description of the experimental model and treatments. Please note that the materials and methods section is at the end of the article. When you start to read the results, you look at figure 1 and you do not know what is CK, HFD+H, etc. because it is not explained in the results or in the figure legend.
In this section they also quantify the NAFLD activity score in the liver, assessing parameters such as encompassing parameters like inflammation, ballooning and steatosis. They should explain in materials and methods how they quantify each of these parameters to arrive at the NAFLD activity score.
2. In the 2.2. Effects of PPE on cholesterol metabolism Section, they measure the expression of ATP binding cassette transporters in the ileum. But how is the expression of ATP binding cassette transporters in the liver? In the liver, ABC transporters are involved in diverse physiological processes including export of cholesterol, bile salts, and metabolic endproducts.
3. In the western-blot of Figures 2 and 4, the bands should be quantified and the values normalised by the corresponding band of the housekeeping protein used. And these normalised values for each protein analysed are represented in a graph with the corresponding error.
4. Page 6 line 175-176, when the authors say “In contrast, the HFD+H group exhibited increased levels of liver CYP7A1, CYP27A1, FXR, SHP, BSEP, and NTCP.” They do not indicate with respect to which experimental group the comparison is made. In this case it is with respect to the HFD, although in some cases it is also with respect to the CK group.
5. In 2.5. Correlation analysis between gut microbiota and BAs metabolism section, a short introduction or explanation of what they have measured and analysed is missing. And then they tell us the results.
When they say "Gut microbiota with BSH activity" which microbiota is this? I think they should indicate it in brackets to make it easier for the reader to understand the results.
They also quantify the ratios DCA/(DCA+CA) and LCA/(LCA + CDCA) without explaining what these ratios indicate.
6. Abbreviations are defined the first time they are used and then it is used throughout the text.
Undefined abbreviations defined CA, CDCA, βCA, βMCA, βMCA, 162 ωMCA, CA-3S, DCA, 23norDCA, LCA, 7,12diketoLCA, THDCA, ASBT
7. In figure legends, all abbreviations used in the figure must be defined. Since the figures are independent of the text, the reader must be able to understand the figure without having to read the text. The authors do not define any or almost none of the abbreviations used in the figures in the figure legends.
8. Discussion, page 10 line 235-236, they said “The intervention with PPE leads to a decrease in liver T-BAs and an increase in fecal T-Bas”. The authors only show data of T-BAs in fecal samples, not in liver. But they also affirm that after the PPE intervention T-BAs increase in fecal samples. If we look at the results shown in figure 3, figure 3a, we can see that the levels of fecal T-BAs in the HFD-H group are lower than in the HFD group. If we look at the individual data for the different BAs, they decrease after treatment with PPE compared to HFD, except for THDCA (Fig. 3D). Therefore, the authors' claims do not correspond to their results.
9. In material and methods section, in 3.2. Animals experimental design, the authors do not indicate whether the protocol has been approved by the relevant Ethical Committee on Animal Experimentation, which oversees animal welfare.
They do not indicate the brand name of the HFD or standard chow diet.
They also do not indicate the route of administration of the PPE or statin.

Reviewer 3 Report
Comments and Suggestions for Authors
The authors present a study that explores the mechanisms of the positive effects of
Prunus cerasifera Ehrh (PPE) on cholesterol and bile acids. The study was performed on a mouse model that was subjected to a continuous high-fat diet and administered PPE. The study shows a promising approach to combat the metabolic dysfunctions,
to treat non-alcoholic fatty liver disease, and improve gut microbiota function. The Work is novel and its publication is recommended once the observations expressed here have been solved.
1.- lines 29-31 – this conclusion is too general. One cannot judge about any plants from arid region other than PPE. Add conclusive line in the end of abstract
2. The abstract is too long.
3. Add quantitative results in the abstract so that visibility and understanding of your article could be better.
4. The abbreviations such as high fat diet (HFD) group, control (CK), etc. should be signed under figures for clarity. All abbreviations should be explained when they first appear.
5. In Introduction it should be explained which polyphenols were found in PPE.
6. Statistical analysis is not clear can you please elaborate values and SD in your results section at least for total cholesterol and BA.
7. In discussion please, mention if there are any data available on this topic in humans.
I am looking forward to receiving an improved version of this manuscript that addresses all these issues .
Best regards
Comments on the Quality of English LanguageThe English language is good. Minor corrections should be carried out.
Reviewer 4 Report
Comments and Suggestions for Authors
REVIEW: MAJOR REVISIONS
In this work, Prunus cerasifera Ehrh. mitigation of the development of non-alcoholic fatty liver disease (NAFLD) through the modulation of cholesterol and bile acids (BAs) metabolism was investigated. The work reported is worthy of investigation, but there are some corrections that need to be done before publication.
- Firstly, the Title should be remodulated in: ‘Therapeutic Role of Polyphenol Extract from Prunus cerasifera Ehrh. on Non-Alcoholic Fatty Liver’. Indeed, the authors cite ‘extracts’, but just one extract was tested.
- Abstract, line 13. ‘Ehrh.,’ Please remove comma.
- The authorship accompanying the plant scientific name should not be written in italics. Moreover, one cited the first time, the scientific name of the plant should be written in its abbreviated form (P. cerasifera).
- Introduction, line 89. More information on the plant should be added in the introduction, supporting its choice for the study. Moreover, the authors cite the plant polyphenols without specifying which are the main polyphenols found in the plant and responsible for its biological activity. In summary, more space should be given to the plant, its traditional uses, its phytocomplex, and its pharmacological properties supporting its choice for the study.
- Materials and methods, Section 3.1. The authors should specify the kind of extraction that was performed, as well as the yield of the extraction and of the subsequent purification.
- The authors declare that the extract is rich in polyphenols, but there are no data supporting this. The authors should add to the manuscript the phytochemical analysis of the extract that was tested in terms of polyphenols content. They could perform some general spectroscopic assays like the TPC or to be more precise and rigorous UHPLC-PRM-MS of the extract. This would enrich the manuscript also from a phytochemical point of view.
Comments on the Quality of English Language
Minor editing of English language required
Round 2
Reviewer 1 Report
Comments and Suggestions for Authors
All the suggestions have been taken into consideration and the missing data have been added.
Reviewer 4 Report
Comments and Suggestions for Authors
The Manuscript has been improved and can be accepted for publication.